# Intraocular Pressure Changes during Hemodiafiltration with Two different Concentrations of Sodium in the Dialysate

**DOI:** 10.3390/biology11010012

**Published:** 2021-12-23

**Authors:** Claudia Lerma, Nadia Saavedra-Fuentes, Jasbeth Ledesma-Gil, Martín Calderón-Juárez, Héctor Pérez-Grovas, Salvador López-Gil

**Affiliations:** 1Departamento de Instrumentación Electromecánica, Instituto Nacional de Cardiología Ignacio Chávez, Mexico City 14080, Mexico; dr.claudialerma@gmail.com (C.L.); martin.calderon@ciencias.unam.mx (M.C.-J.); 2Departamento de Nefrología, Instituto Nacional de Cardiología Ignacio Chávez, Mexico City 14080, Mexico; nari40_2@hotmail.com (N.S.-F.); hpgrovas@gmail.com (H.P.-G.); 3Departamento de Glaucoma, Instituto de Oftalmología Conde de la Valenciana, Mexico City 06800, Mexico; dra.jasbeth.ledesma@gmail.com; 4Plan de Estudios Combinados en Medicina, Facultad de Medicina, Universidad Nacional Autónoma de México, Mexico City 04510, Mexico

**Keywords:** intraocular pressure, hemodiafiltration, sodium concentration, intra-dialytic changes

## Abstract

**Simple Summary:**

An increase in intraocular pressure during chronic hemodialysis is linked to ocular complications, such as glaucoma. The behavior of intraocular pressure during hemodiafiltration is unknown. Changes in intraocular pressure with a sodium dialysate concentration fixed at 138 mmol/L and an individualized concentration were studied in 13 patients with end-stage renal disease treated with hemodiafiltration. Up to 31% patients presented an episode of intraocular hypertension without differences between sodium profiles. A large variability in intraocular pressure within patients and a high prevalence of transient intraocular hypertension were found.

**Abstract:**

Ocular complications are common among end-stage renal disease patients and some complications had been linked to increments of intraocular pressure (IOP) during hemodialysis. The changes of IOP during hemodiafiltration (HDF) have been scarcely investigated and the potential influence of the sodium dialysate concentration is unknown. The aim of this study was to compare the IOP changes during HDF with sodium dialysate concentration, either fixed or individualized. Thirteen end-stage renal disease patients participated in the study; they were treated with HDF using a dialysate sodium profile fixed at 138 mmol and another session with an individualized sodium profile. The intraocular pressure was measured before and after each session and every 30 min during HDF. Both groups had a similar HDF prescription, blood pressure, and biochemical parameters. At the end of hemodiafiltration, sodium concentration decreased only in the fixed sodium profile group. The number of patients with at least an episode of intraocular hypertension during HDF ranged from 5 (19%) to 8 (31%) without significant differences between right and left eye nor between dialysate sodium concentration. During HDF, there is a large variability of IOP; transient events of intraocular hypertension are highly prevalent in this sample, and they are not related to the sodium dialysate concentration.

## 1. Introduction

End-stage renal disease is a major public health problem in industrialized and developing countries, it is highly prevalent and requires expensive therapies to maintain patients’ long-term survival [1,2]. There are several ocular complications in advanced kidney disease and hemodialysis, such as: red eye, retinal hemorrhage, macular leakage, optic neuropathy, band keratopathy, and high intraocular pressure (IOP) [3,4]. Elevated IOP pressure is a crucial risk factor for the development of glaucoma, even if it is within normal pressure range [5,6].

In healthy individuals, an increase in salt consumption is associated to a discrete decrease on IOP [7]. Moreover, a significant reduction of IOP has been seen after 18 h of dehydration in healthy volunteers [8]. The administration of intravenous hypertonic saline decreases IOP in a short-term period in patients with glaucoma [9].

There are several case reports of important IOP increments, glaucoma and retro-ocular pain during hemodialysis [10,11,12,13,14,15,16,17]. The IOP narrow balance relies on the production of aqueous humor by the ciliary body and its outflow through the trabecular meshwork and the inner wall of Schlemm’s canal [18]. The composition and osmolarity of the aqueous humor are similar to the plasma; nevertheless, it has a lower protein concentration and a lower concentration of glucose and urea [19]. It is thought that an electrolyte and osmolarity disequilibrium might lead to a rise in IOP, since with the reductions in body fluid volume and osmotic pressure caused by hemodialysis (HD), the amount of aqueous humor declines mainly by changes in intraocular osmotic pressure rather than plasma osmolality [20].

In a previous study, it was shown that the IOP during HD and hemodiafiltration (HDF) has a similar behavior [21]. However, there are no studies describing the changes of IOP during HDF and the potential influence of the sodium dialysate concentration. The aim of this study was to compare the IOP changes during HDF with sodium dialysate concentration, either fixed or individualized.

## 2. Materials and Methods

### 2.1. Study Participants

This exploratory study included 13 adult patients with end-stage renal disease, under treatment with HDF three times per week for at least 3 months. Patients were recruited consecutively (non-random sampling). Inclusion criteria were male or female adult patients. Exclusion criteria were previous symptoms of retro-ocular pain, previously diagnosed glaucoma, newly diagnosed glaucoma (ophthalmological assessment is described below) or use of ocular hypotensive drugs. Elimination criteria were patients that showed hemodynamic instability during hemodialysis (either by clinical symptoms or hypotensive blood pressure values).

All patients underwent ophthalmological bio-microscopy and fundoscopy by an ophthalmologist. IOP was measured in both eyes under topical anesthesia with tetracaine 0.5% eyedrops in both eyes. All procedures performed in studies involving human participants were in accordance with the ethical standards of the Research and Ethics Committee of the Instituto Nacional de Cardiología Ignacio Chávez (protocol number 14-891) and with the 1964 Helsinki declaration and its later amendments. Informed consent was obtained from all the participants.

### 2.2. Hemodialysis Prescription

HDF sessions were delivered by volumetric hemodialysis machines (FMC-4008H, Fresenius Medical Care, Bad Homburg, Germany). Ultrapure dialysate (mmol/L): HCO_3_^−^ = 35, Na^+^ = 138, K^+^ = 2; (mEq/L) Ca^2+^ = 3.5, Mg^2+^ = 1, and polysulfone membranes were used (F-80, Fresenius Medical Care, Walnut Creek, CA, USA). Two dialysate solutions were used, each with a different sodium’s concentration. Each patient received one HDF session with a profile of dialysate sodium concentration fixed at 138 mmol/L (standard sodium concentration in our center) and another HDF session with individualized sodium. The personalized profile started the HDF session with a dialysate sodium concentration was set to the same value of the venous sodium concentration and gradually moved towards the standard concentration during HDF (one mEq/L every 36 min). Every session lasted 3 h with fixed ultrafiltration profile and HDF were similar in both sessions (Table 1). All patients had indications for a non-restricted diet, did not use erythropoietin, and participated in a program or aerobic exercise (cycling in recumbent position with modified bicycles) during all sessions.

### 2.3. Study Protocol and Intra-Ocular Pressure Assessment

Before the dialysis started, a venous blood sample was obtained in order to measure biochemical parameters (glucose, blood urea nitrogen, creatinine, chlorine, sodium, potassium, calcium). The following dialysis parameters were recorded at 30, 60, 90, 120, 150, and 180 min during HDF and 30 min after the end of the session: systolic and diastolic blood pressure, heart rate, temperature, and oxygen saturation level. At the end of the session, a blood sample was obtained from the venous access to obtain the same parameters as in the first blood sample and the following dialysis parameters were recorded: total filtration volume (mL), ultrafiltration rate (mL/h), blood flow rate (mL/min), arterial pressure (mmHg), venous pressure (mmHg) and final body weight (Kg). The osmolality was estimated with the following formulae: osmolality = 2 × (Na^+^ + K^+^) + ([glucose]/18) + ([blood urea nitrogen]/2.8).

IOP was evaluated with a tonometer (TonoPen, model AVIA, Reichert Technologies, Depew, NY, USA). Baseline IOP was measured before the beginning of dialysis. During dialysis, intraocular pressure was evaluated in millimeters of mercury (mmHg) at the following times: 30, 60, 90, 120, 150, 180 min, and after the end of the session. Intraocular hypertension episode was defined as an intraocular pressure ≥22 mmHg measured during hemodialysis.

### 2.4. Statistical Analysis

Kolmogorov–Smirnov tests were applied to quantitative variables in order to test for normal distribution. The results of the variables with a normal distribution are reported as mean and standard deviation and were compared using *t* test (two group comparisons) or analysis of variance for repeated samples with post-hoc analysis using the Bonferroni method (3 or more groups). The variables without a normal distribution are reported as median (percentile 25–percentile 75) and were compared with the U Mann–Whitney test. The absolute value and percentage were used to report qualitative variables and were compared with Chi-squared test or Exact Fisher’s test. The statistical analysis was performed with SPSS version 15.0 (IBM). Considering that there are no previous reports on the effect of the sodium dialysate profile tested in this study, we could not perform an *a priori* sample size estimation. However, we estimated an achieved statistical power of 0.56 with the results of the present exploratory study (See detailed description in Appendix A).

## 3. Results

The study included 13 patients (eight men and five women), with mean age 42 ± 7 years old, median HD vintage was 191 (80–436) weeks. End-stage renal disease etiology was systemic lupus erythematosus (*n* = 2), urate nephropathy (*n* = 3), IgA nephropathy (*n* = 2), kidney-transplant rejection (*n* = 4), focal segmental glomerulosclerosis (*n* = 1), or collapsing glomerulopathy (*n* = 1). The hemoglobin measured by routinely laboratory measurements in both groups before HDF sessions was 9.7± 3.1 g/dL and after HDF sessions was 12 ± 3.2 g/dL (*p* < 0.01). The vascular access was arteriovenous fistula (*n* = 7) or central venous catheter (*n* = 6).

The characteristics of two hemodiafiltration sessions with different profiles of dialysate sodium concentration were similar in both groups (Table 1). In the individualized dialysate sodium HDF profile, the sodium concentration remained similar before and after the HDF session (Figure 1). The sodium concentration after the HDF session with individualized profile is greater than the sodium concentration after the HDF session with a fixed profile, although the sodium concentration goal was achieved in both groups. The blood lactate remained similar before and after the fixed HDF session, but it increased after the individualized sodium profile. There was no difference in blood glucose at the beginning and at the end of the HDF session in any of the groups. It is remarkable that the osmolality at the end of the HDF in fixed sodium profile is significantly lower compared with the beginning of the session. Similar changes in potassium, pH and bicarbonate were observed in both sessions.

Among patients, there is a great variability in the IOP values along HDF (Figure 2). At 120, 150 and 180 min, in both groups there were more positive IOP changes (IOP change ≥ 1 mmHg) than negative changes (Table 2). At 150 and 180 min, in both groups there were more positive IOP changes (IOP change ≥ 6 mmHg) than negative changes.

There is a high number of intraocular hypertension events along HDF with no difference between groups (Table 3). The number of patients with at least an episode of intraocular hypertension during HDF ranged from five (19%) to eight (31%) without significant differences between right and left eye nor between dialysate sodium concentration.

Figure 3 shows the mean values of IOP change during hemodiafiltration. There were no significant differences on IOP between sessions. In the group with a fixed profile at 60 min, there was an increase in IOP of the right eye compared with the IOP of the same eye at 30 min. At 30 min in the fixed sodium concentration group there was an increase in IOP of the right eye compared with the individualized sodium HDF profile of the same eye.

Although, there was no difference in blood pressure, there was a decrease in heart rate at 30, 60, 90, 120, and 150 min in the fixed sodium profile HDF group compared with its own baseline at the beginning of HDF (Figure 4).

## 4. Discussion

HDF has different characteristics compared to hemodialysis. HDF clears larger toxins and provides some clinical advantages, such as better control of hyperphosphatemia, hemodynamic stability and control of fluid overload [22]. In spite of the advantages, elevated IOP might occur. HDF introduces several modifications into cardiovascular functioning and concentration of dissolved substances in the blood.

Increased IOP during renal replacement therapy is a latent problem. Some strategies have been proposed to prevent these changes, for instance, the use of acetazolamide [23], intravenous hyperosmotic solutions [24], mannitol [25], and glucose [16]. We evaluated the effect of maintaining a relatively high concentration of sodium in order to reduce the tendency of the IOP to increase.

In the present study, we found variations in heart rate during HDF with sodium dialysate fixed at 138 mmol/L. It is known that an increase in osmolarity leads to an increase in heart rate, probably due to autonomic nervous regulation [26]. Despite the variation in heart rate, we did not find a significant change in blood pressure; as mentioned above, HDF provides a better hemodynamic management [27].

In this study, a large variability in IOP through the HDF session was observed among patients. This phenomenon reflects the complexity of IOP regulation. Although there is a significant increase in IOP of the right eye compared with the left eye at 30 and 60 min of the HDF session, we observed a tendency of IOP positive change events (IOP change ≥ 1 mmHg) and the severity of these changes (IOP change ≥6 mmHg) to increase towards the end of the HDF in both groups. Despite a significantly lower osmolality in the fixed sodium profile at the end of the HDF compared with the beginning of it, there was no difference compared with the individualized sodium profile and the IOP values. Although the results did show a significant difference of sodium after dialysis (137.7 ± 1.3 vs. 138.9 ± 2.5 mEq/L), this difference might be too small to cause any difference of IOP. Probably the variation in IOP relies with much more importance in the production of aqueous humor rather than the outflow from the eye.

A previous study found that increased blood glucose levels decrease IOP in hemodialysis [28]. However, other mechanisms are involved in the increase of IOP, for example the anterior chamber angle wideness [29], aqueous flow, and outflow resistance [30].

IOP is given by the flow rate of the aqueous humor that drains in the trabecular meshwork into the Schlemm canal, the input rate of aqueous flow and the episcleral venous pressure. The modification in any of these variables will modify IOP. Whether the output of aqueous flow rate decreases or the input flow rate or the episcleral venous pressure increases, the IOP will also increase [31]. Although the pathophysiology of glaucoma is not fully understood, there is an association between elevated IOP and glaucoma [32,33].

During HDF, the body fluid volume, plasma osmolarity, and electrolyte concentration is corrected [34]. It has been reported that plasma osmolarity is related with IOP [35], thus it was hypothesized that a sharper correction of sodium (the main electrolyte contributing to osmolarity) may decrease the gradient between plasma and interstitial fluid and promote a greater input on aqueous flow.

The transient increase of IOP during HDF may be subject to a therapeutic intervention as part of the management of patients during the HDF session. However, the changes are rather heterogenous in hemodialysis [36,37], for example, a decrease of IOP has been reported during hemodialysis [38], while we found changes consisting of both increases and decreases in IOP. Whether these fluctuations are related to glaucomatous progression in these patients, remains unknown. The potential relation of increased IOP in hemodialysis and HDF with adverse clinical outcomes should be evaluated with other study designs, such as cohorts that include patients with and without glaucoma.

The measuring of IOP is a technical challenge, although applanation tonometry is the most accurate method, it requires the patient to sit and mount a slit lamp inside the hemodialysis center. Furthermore, changes in the position of the patient are routinely made as part of hypotension management, which is a difficulty for the application of applanation tonometry [39]. Tonopen provides an acceptable measurement of IOP, though the measurement of applanation and Tonopen tonometry values are not necessarily interchangeable [40]. For a continuous measurement of IOP, other techniques, such as rebound tonometer [41] and pressure-measurement contact lens [42], may be considered in future studies. Tonopen AVIA, used in this work, provides reliable measurements in cases without access to a slit-lamp [43]. Nevertheless, other novel tonometers should be considered for future projects [44].

An important advantage in this study is that the same individuals received both sodium profiles with similar HDF prescriptions; therefore, there was a better control of variability among individuals. Another advantage is the frequency of intra-dialysis IOP measurements, which is likely to allow the identification of transient IOP changes. The limitations in this study include the small number of participants (which resulted in a low achieved statistical power) and the measurement of the data from only one HDF session per group. The limited number of volunteers in this study compromises the external validity of our findings, which should be corroborated with further clinical research. On the grounds that there is a great variability in IOP during hemodiafiltration, IOP measurement should be taken into consideration in patients with risk factors of developing glaucoma. It is known that there is a diurnal variation of IOP intraocular pressure, the mean range of fluctuation is 5 mmHg [45]. Therefore, the increase in IOP greater than this value cannot be explained by the diurnal variation.

## 5. Conclusions

There is a large variability of IOP during HDF. Transient events of intraocular hypertension are highly prevalent during this HDF pilot study and they are not related to sodium concentration nor osmolarity. The results of this exploratory study highlight the need for more pertinent diagnostic and treatment strategies to prevent further visual field damage in dialysis patients.

## Figures and Tables

**Figure 1 biology-11-00012-f001:**
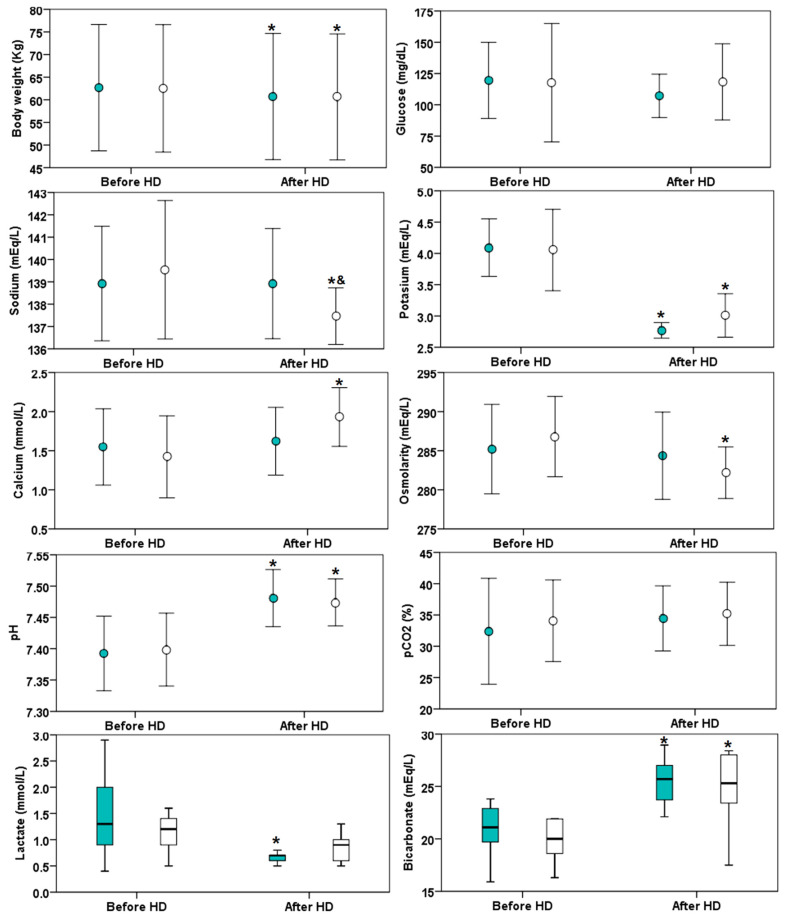
Biochemical parameters before and after hemodiafiltration (HDF) in 13 patients with different dialysate sodium profile: individualized indicated (green color markers) or fixed at 138 mmol/L (white markers). The asterisk (*) indicates *p* < 0.05 versus before HDF and the symbol & indicates *p* < 0.05 versus fixed sodium. The circle markers indicate mean values (± standard deviation) while the box plots indicate medians (horizontal line), interquartile range (box) and 1.5 time the interquartile range (error bars).

**Figure 2 biology-11-00012-f002:**
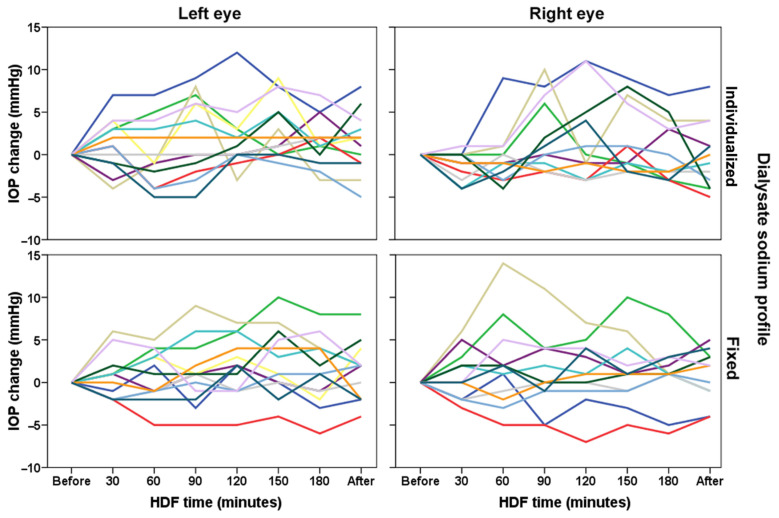
Individual values of intraocular pressure evaluated during hemodiafiltration (HDF) in 13 patients (each color indicates a different patient) with different dialysate sodium profile (individualized or fixed at 138 mmol/L).

**Figure 3 biology-11-00012-f003:**
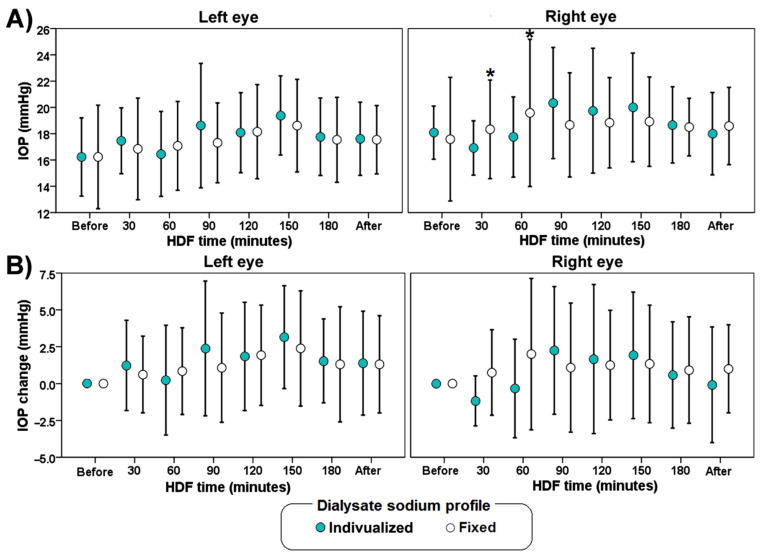
Average values of IOP (**A**) and IOP change (**B**) during hemodiafiltration (HDF) with different dialysate sodium profile (individualized or fixed at 138 mmol/L). The error bars indicate one standard deviation. The asterisk (*) indicates a significant difference (*p* < 0.05) versus measurements in the left eye.

**Figure 4 biology-11-00012-f004:**
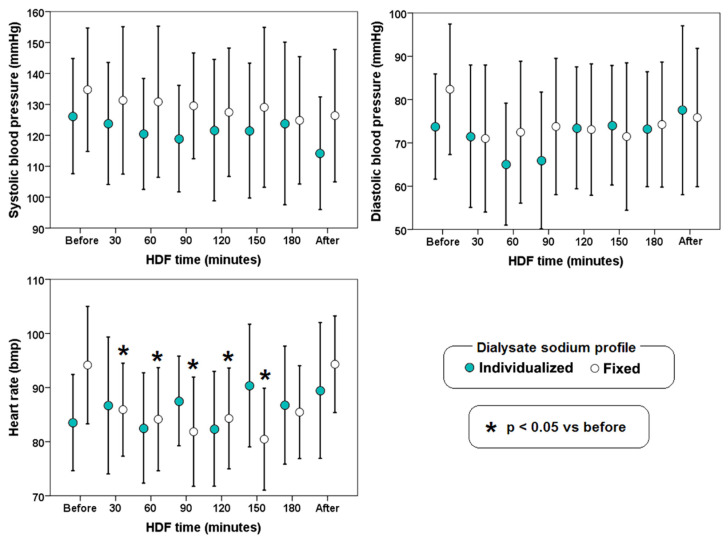
Blood pressure and heart rate during hemodiafiltration (HDF) with different dialysate sodium profiles (individualized or fixed at 138 mmol/L). The error bars indicate one standard deviation.

**Table 1 biology-11-00012-t001:** Characteristics of two hemodiafiltration sessions with different profile of dialysate sodium concentration applied in 13 patients. The results are reported as mean ± standard deviation or median (percentile 25–percentile 75).

	Profile of Dialysate Sodium Concentration		
	Fixed	Individualized	*p*-Value
HDF session time (minutes)	180 (180–210)	210 (180–210)	0.15
Total ultrafiltration volume (mL)	2000 (1200–2500)	2200 (2000–2462)	0.07
Blood flow (mL/min)	450 ± 46	454 ± 42	0.31
Ultrafiltration rate (mL/h)	666 (399–668)	657 (565–684)	0.63
Dialysate sodium (mEq/L)	138 (138–138)	140 (137–141)	0.26

**Table 2 biology-11-00012-t002:** Number of episodes with changes in IOP compared to pre-HDF.

Time (min)	Change ≥ 1mmHg			Change ≥ 6 mmHg		
	Total	Increase	Decrease	Total	Increase	Decrease
30	40 (80%)	21 (53%)	19 (48%)	3 (6%)	3 (6%)	0 (0%)
60	47 (94%)	23 (49%)	24 (51%)	3 (6%)	3 (6%)	0 (0%)
90	44 (88%)	26 (59%)	18 (41%)	12 (24%)	12 (24%)	0 (0%) **
120	44 (88%)	29 (66%)	15 (34%) *	8 (16%)	7 (14%)	1 (2%) *
150	44 (88%)	31 (70%)	13 (30%) **	12 (24%)	12 (24%)	0 (0%) **
180	47 (94%)	32 (68%)	15 (32%) *	7 (14%)	5 (10%)	2 (4%)

* *p* < 0.05 (increase vs. decrease) ** *p* < 0.01 (increase vs. decrease).

**Table 3 biology-11-00012-t003:** Events of IOP ≥ 22 mmHg in 26 pairs of eyes (13 patients).

Dialysate Sodium Concentration	Time of Evaluation during Dialysis (min)							
	Before	30	60	90	120	150	180	After
Fixed	2	3	4	4	3	4	1	2
	(7.6%)	(11.5%)	(15.3%)	(15.3%)	(11.5%)	(15.3%)	(3.8%)	(7.6%)
Individualized	2	1	2	3	3	6	4	1
	(7.6%)	(3.8%)	(7.6%)	(11.5%)	(11.5%)	(23%)	(15.3%)	(3.8%)

## Data Availability

Data is available upon reasonable request.

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
