# Peer review of "Intraocular Pressure Changes during Hemodiafiltration with Two different Concentrations of Sodium in the Dialysate"

_biology, 2021, doi:10.3390/biology11010012_

Round 1
Reviewer 1 Report
Lerma at el present a manuscript where they analyzed Intraocular pressure (IOP) changes in end stage renal disease patients who received hemodialysis. The authors manipulate the sodium profile in the hemodialysate to determine if that affects IOP changes and they find that there are IOP changes in these patients but unrelated to the sodium profile. The small number of participants in this study makes it difficult to draw strong conclusions from the dataset as the authors themselves realize. It is understandable that it may be challenging to find many volunteers with end stage renal disease to participate in the study.
1.Have the authors done any power analysis to determine that their sample size holds sufficient statistical power?
2. The authors interpretation that there are large changes in IOP could be a bit overstated because of the small samples size.
3. The authors find that IOP changes that they observed during hemodialysis is not related to the factor that they manipulated , which leads to questions about their methodology and hypothesis. Would varying sodium concentration affect IOP in healthy individuals? Is this well described? Are there other aspects of hemodiafiltration that could be contributing to IOP changes and should these be explored rather than sodium concentration?
4. Data presentation in some contexts is very hard to digest. Examples table 1, 2 , 3 and 4 would have been a lot easier to digest if they were presented in a graphical format rather than a table.
Reviewer 2 Report
The authors presented an interesting study comparing the changes in intraocular pressure among end‐stage renal disease patients during hemodiafiltration with sodium dialysate concentration either fixed or individualized. The manuscript is with merit and the findings are worth reporting. However, before publication could be considered, the authors should revise the manuscript and address the following concerns.
- The information about the details regarding the enrolled subjects (gender, age, etiology of the end‐stage chronic renal disease etc (lines 62-72) should be moved from this section “2.1 Study participants” to the beginning of the “Results” section. Lines 74-78 should be kept in this section with additional detailed information about the criteria of inclusion and exclusion of the study participants.
- Line 72: the detailed information about the “complete assessment by an ophthalmologist” should be provided. Detailed information should be provided regarding the method of measurement of IOP. The authors discuss in the results section the occurrence of episodes of “intraocular hypertension”: in the methods the specific details about the IOP measurements and definitions should be provided
- Lines 94-97: the information should be moved from this section “2.2” to the “Results” section.
- Statistics: The author should provide a statistical power estimation for their study or at least some justification of the study n and add it in the section of the methods entitled “Statistical analysis”
- The authors should expand their discussion section and provide some additional insights about the clinical relevance of their research.
Reviewer 3 Report
The manuscript entitled “Intraocular pressure changes during hemodiafiltration with two different concentrations of sodium in the dialysate” is a study based on the effects of dialysis on intraocular pressure. The study is of clinical interest and adds to the limited literature in this field. Specialized clinicians tend to be sectorial in treatment, and studies like the one presented here remind us to treat the patient as a person with complex physiopathological interconnected processes and not simply as a collection of independent organs.
The authors present a good study plan and provide interesting clinical data. Statistical analysis considered are appropriate. The tables and figures are pertinent and useful in summarizing the main issues reported in the text.
There are minor comments that can be raised. Additional information should be included in the Introduction and Discussion sections to better describe the mechanisms involved that could influence intraocular pressure changes. This could be of clinical use to understand the underlying pathways in normal and glaucomatous patients with renal disease and provide insights to possible therapeutic prevention measure that could be considered prior to hemodialysis in glaucomatous patients at risk of damage and/or glaucomatous progression.
Tonopen was used to measure IOP, which is an appropriate instrument for this type of study. Brief mention with appropriate references should be added in the Discussion section to explain why gold standard Goldmann tonometry was not used (besides the obvious need for slit-lamp microscopy) or other new alternatives like iCare tonometry, which could be an interesting option in future studies.
The authors present preliminary data. A note should be made with regards to limitations and future studies that should include a larger cohort, normal versus glaucomatous patients, with inclusion of severity data for glaucomatous patients. The results of this type of future study could prove to be important in patients that have end-stage glaucoma that require continual hemodialysis, in that more stringent treatment strategies and lower target pressures could be necessary to prevent further visual field damage if transient intraocular pressure peaks prove to be of clinical importance in these patients.
Round 2
Reviewer 2 Report
The authors addressed the comments to the best of their ability and the manuscript can be accepted for publication.